# Association of Pulse Volume Recording at Ankle with Total and Cardiovascular Mortality in Hemodialysis Patients

**DOI:** 10.3390/jcm8122045

**Published:** 2019-11-21

**Authors:** Wen-Hsien Lee, Po-Chao Hsu, Jiun-Chi Huang, Ying-Chih Chen, Szu-Chia Chen, Pei-Yu Wu, Meng-Kuang Lee, Chee-Siong Lee, Hsueh-Wei Yen, Ho-Ming Su

**Affiliations:** 1Graduate Institute of Clinical Medicine, College of Medicine, Kaohsiung Medical University, Kaohsiung 807, Taiwan; cooky-kmu@yahoo.com.tw (W.-H.L.); karajan77@gmail.com (J.-C.H.); 2Department of Internal Medicine, Kaohsiung Municipal Siaogang Hospital, Kaohsiung 812, Taiwan; 990329kmuh@gmail.com (Y.-C.C.); scarchenone@yahoo.com.tw (S.-C.C.); 980261kmuh@gmail.com (M.-K.L.); 3Faculty of Medicine, College of Medicine, Kaohsiung Medical University, Kaohsiung 807, Taiwan; pochao.hsu@gmail.com (P.-C.H.); lcsphk@ms18.hinet.net (C.-S.L.); hweyen@cc.kmu.edu.tw (H.-W.Y.); 4Division of Cardiology, Department of Internal Medicine, Kaohsiung Medical University Hospital, Kaohsiung 807, Taiwan; 5Division of Nephrology, Department of Internal Medicine, Kaohsiung Medical University Hospital, Kaohsiung 807, Taiwan; wpuw17@gmail.com

**Keywords:** mortality, percent of mean arterial pressure, upstroke time, pulse volume recording

## Abstract

Pulse volume recording is an accurate modality for detecting arterial occlusion in the lower extremities. There are two indexes of pulse volume recording measured at ankle, percentage of mean arterial pressure (%MAP) and upstroke time (UT). The aim of the study was to examine the ability of %MAP and UT for the prediction of overall and cardiovascular mortality in hemodialysis (HD) patients. In 197 routine HD patients, ankle %MAP, ankle UT, and ankle–brachial index (ABI) were automatically measured by Colin VP-1000 instrument. Fourteen cardiovascular mortality and 29 overall mortalities were documented during 2.7 ± 0.6 years follow-up. In the univariate analysis, in addition to co-morbidities and traditional clinical parameters, increased total mortality was associated with decreased ABI, ABI < 0.9, increased %MAP and UT, %MAP > 50%, and UT > 169 ms (*p* ≤ 0.041) and increased cardiovascular mortality was associated with increased UT and %MAP > 50% (*p* ≤ 0.022). After multivariate analysis, increased %MAP and %MAP > 50% (*p* ≤ 0.047) were still the predictors of total mortality and %MAP > 50% (*p* = 0.024) was still the predictor of cardiovascular mortality. In HD patients, we found that ankle %MAP and %MAP > 50% could predict total mortality and ankle %MAP > 50% could predict cardiovascular mortality in the multivariate analysis. Hence, assessment of %MAP from pulse volume recording at ankle might be helpful in identifying the high-risk group for poor prognosis in HD patients.

## 1. Introduction

The prevalence of peripheral artery disease (PAD) has increased in recent years. The presence of PAD impairs quality of life and is associated with a greatly increased risk of major cardiovascular events and death [1]. The risk factors of PAD are old age, the presence of hypertension and diabetes, smoking, and renal insufficiency [2,3]. Among patients with end-stage renal disease on hemodialysis (HD), the prevalence of PAD is high, ranging from 10% to 35% [4,5].

Pulse volume recording is another accurate modality for detecting arterial occlusion in the lower extremities [6]. There are two indexes of pulse volume recording measured at ankle, percentage of mean arterial pressure (%MAP) and upstroke time (UT). The two indexes are useful parameters for the diagnosis of PAD in patients with normal ankle brachial index (ABI) [7]. The %MAP can predict all-cause mortality in patients with normal ABI during 20.3 months of follow-up [8]. Compared with ABI, UT showed a significantly stronger association with vascular and renal damage in an elderly Chinese cohort [9]. In addition, UT was positively associated with cardiovascular outcomes in patients with type 2 diabetes [10] and could predict all-cause and cardiovascular mortality in the Chinese elderly [11]. However, there is no study to evaluate the association between pulse volume recording measured at ankle and mortality in HD patients, a group with an extraordinarily high prevalence of PAD. Hence, the aim of the present study was to examine the ability of %MAP and UT measured at ankle for the prediction of overall and cardiovascular mortality in HD patients. In addition, we also assessed the major determinants of %MAP and UT in such patients.

## 2. Methods

### 2.1. Study Patients and Design

The study was performed in a regional hospital in southern Taiwan from April 2016. All routine HD patients in the dialysis clinic of this hospital were included except those who refused vascular examination (*n* = 6) and those with atrial fibrillation (*n* = 4). Finally, 197 patients were included in this study. The protocol was approved by our Institutional Review Board and all enrolled patients gave written, informed consent.

### 2.2. Hemodialysis

All patients received routine HD three times per week using a Toray 321 machine (Toray Medical Company, Tokyo, Japan). Every HD session was conducted for 3–4 h using a dialyzer with a blood flow rate of 250 to 300 mL/min and dialysate flow of 500 mL/min.

### 2.3. Measurements of Peripheral Vascular Parameters and Blood Pressures

After resting in the supine position for at least 5 min, the peripheral vascular parameters and blood pressures were measured 10–30 min before HD using a validated device (VP-1000; Colin Corporation, Hayashi, Komaki City, Japan) that automatically and simultaneously measures blood pressures in both arms and ankles using an oscillometric method [12]. Occlusion and monitoring cuffs were placed tightly around the upper arm without blood access and both sides of the lower extremities. Measurement details on brachial–ankle pulse wave velocity (baPWV) and ABI were mentioned in our previous study [13]. In brief, the ABI was counted by the ankle systolic blood pressure divided by the higher arm systolic blood pressure. The baPWV was calculated as the transmitted distance from the pulse wave from brachial to tibial arteries divided by passage time of the pule wave. The %MAP and UT were automatically determined based on the pulse volume recordings of ankle. The %MAP was calculated as the height of the mean area of the arterial wave divided by the peak amplitude. The UT was calculated as the time interval from the onset to the peak of a pulse volume wave. After obtaining bilateral values, we chose the higher baPWV, %MAP, and UT and the lower ABI for later analysis. All the above measurements were performed once in each patient.

### 2.4. Collection of Demographic, Medical, and Laboratory Data

Demographic and medical data including age, gender, current smoking history, and comorbid conditions were acquired from medical records and interviews with patients. The body mass index (BMI) was calculated as the ratio of weight in kilograms divided by the square of height in meters. Blood samples were obtained within 1 month of enrollment. The diagnosis of diabetes mellitus was confirmed if the fasting blood glucose level was greater than 126 mg/dL or hypoglycemic agents were used to control blood glucose. The diagnosis of hypertension was confirmed if the systolic blood pressure was ≧140 mmHg or diastolic blood pressure ≧ 90 mmHg or anti-hypertensive agents were prescribed. Stroke was defined as a history of cerebrovascular disease including cerebral bleeding and infarction. Coronary artery disease was defined as angiographically documented coronary artery disease, a history of myocardial infarction, a history of typical chest pain with positive stress test and having undergone coronary artery bypass surgery or angioplasty. Heart failure was defined according to Framingham criteria [14].

### 2.5. Definition of Cardiovascular Mortality

Cardiovascular mortality was defined as death caused by cardiogenic shock, heart failure, ischemia heart disease, lethal arrhythmia, unexplained sudden cardiac death, aortic dissection, cerebrovascular disease, and so on. Cardiovascular mortality was ascertained and adjudicated by two cardiologists, with disagreement resolved by adjudication from a third cardiologist from the hospital course and medical record. In mortality patients, they were followed until date of death. The other patients were followed until March 2019.

### 2.6. Statistical Analysis

SPSS 22.0 software (SPSS, Chicago, IL, USA) was used for statistical analysis. Data were expressed as the mean ± standard deviation or percentage. Continuous and categorical variables between groups were compared by independent samples *t* test and Chi-square test, respectively. The significant variables in the univariate analysis were selected for multivariate analysis. Time to mortality events was modeled using the Cox proportional forward hazards model. A Kaplan–Meier survival plot was calculated from baseline to time of mortality events and compared using the log-rank test. All tests were two-sided and the level of significance was established as *p* < 0.05.

## 3. Results

### 3.1. Baseline Characteristics among Study Patients

Among the 197 subjects, the mean age was 61 ± 12 years. The prevalence of ABI < 0.9 was 25.4%. Table 1 compares the baseline characteristics between patients with and without mortality. Compared to patients without mortality, patients with mortality were found to have an older age, higher prevalence of diabetes, coronary artery disease, stroke, and chronic heart failure, lower albumin, lower ABI, higher UT and %MAP, and higher prevalence of ABI < 0.9, UT > 169 ms, and %MAP > 50%.

### 3.2. Univariate and Multivariate Correlations of %MAP

Table 2 shows the univariate and multivariate correlates of %MAP in study patients. After multivariate analysis, %MAP was positively correlated with baPWV and UT and negatively with male gender and ABI. Table 3 shows the univariate and multivariate correlates of UT in study patients. After multivariate analysis, UT was positively correlated with %MAP and negatively with diastolic blood pressure and ABI.

### 3.3. Kaplan–Meier Analyses of Overall Mortality-Free Survival in Study Patients

The follow-up period to mortality was 2.7 ± 0.6 years in all patients. Mortality events were documented during the follow-up period, including cardiovascular mortality (*n* = 14) and overall mortality (*n* = 29).

The optimal cut-off vales of %MAP and UT for the prediction of overall mortality have not been established. To find the appropriate cut-off values of %MAP and UT as predictors of overall mortality, we created several models using different cut-off values of %MAP and UT. Using the Chi-square value to select the model with the best performance, the model using %MAP > 50% and UT > 169 ms had the best performance in predicting the overall mortality. There were 24 and 99 patients with %MAP > 50% and UT > 169 ms, respectively. Figure 1 illustrates the Kaplan–Meier curves for overall mortality-free survival in study patients subdivided according to %MAP > 50% or not (log-rank *p* = 0.002) and UT > 169 ms or not (log-rank *p* = 0.030). Participants with %MAP > 50% and UT > 169 ms had higher overall mortality rate.

### 3.4. Major Predictors of Overall and Cardiovascular Mortality in Study Patients

Table 4 shows the predictors of total and cardiovascular mortality using the Cox proportional hazards model in the univariate analysis. Increased total mortality was associated with increased age, diabetes, coronary artery disease, stroke, chronic heart failure, decreased albumin, total cholesterol, and ABI, ABI < 0.9, UT > 169 ms, %MAP > 50%, and increased UT and %MAP. Increased cardiovascular mortality was associated with increased age, diabetes, coronary artery disease, stroke, decreased albumin and total cholesterol, increased UT, and %MAP > 50%. The predictors of total and cardiovascular mortality using the Cox proportional forward hazards model in the multivariate analysis is shown in Table 5. The covariates in the multivariate analysis included significant variables in the univariate analysis in Table 4. After adjustment, increased %MAP and %MAP > 50% were still the predictors of total mortality and %MAP > 50% was still the predictor of cardiovascular mortality.

## 4. Discussion

This study aimed to evaluate %MAP and UT measured at ankle for the prediction of overall and CV mortality in HD patients. We found that %MAP and %MAP > 50% were the predictors of total mortality and %MAP > 50% was the predictor of cardiovascular mortality in the multivariate analysis. In addition, after multivariate adjustment, the major determinants of %MAP were female gender, low ABI, high baPWV, and high UT and the major determinants of UT were low diastolic blood pressure, low ABI, and high %MAP in HD patients.

In patients without PAD, pulse volume recording looks like a normal arterial wave. In contrast, in patients with peripheral arterial occlusion, the pulse waveform becomes flattened and has a delayed upstroke. A high %MAP caused by a flattened arterial wave implies the possibility of peripheral arterial occlusion [15]. In fact, our result showed that %MAP had a significantly positive correlation with the presence of ABI < 0.9. Since peripheral arterial occlusion is associated with increased mortality [16], a high ankle %MAP may be able to predict mortality. Our present study actually showed that %MAP was a predictor of total mortality and %MAP > 50% was a predictor of total and cardiovascular mortality in HD patients. In addition, although ABI < 0.9 was a good indicator of peripheral arterial occlusive disease, AbuRahma et al. found that among symptomatic patients with PAD with 50% or greater stenosis on duplex ultrasound examination, 43% had normal resting ABI [17]. Ankle %MAP was reported to be a useful parameter for the diagnosis of peripheral artery disease in patients with normal ABI [7]. Li et al. demonstrated a high %MAP based on pulse volume recording in participants with 0.9 < ABI ≤ 1.3 could predict all-cause mortality during 20.3 months of follow-up [8]. Hence, in patients with normal ABI, %MAP was still a useful predictor of PAD and overall mortality. Our present study demonstrated that even after adjusting ABI and ABI < 0.9, increased %MAP was still associated with total mortality and %MAP > 50% was still associated with total and cardiovascular mortality in patients with HD. Therefore, additional assessment of ankle %MAP might be helpful in the identification of HD patients with high mortality.

Mitsutake et al. found that UT was significantly correlated with coronary artery calcification score based on computed tomography findings [18]. Sheng et al. reported UT per cardiac cycle could predict total and cardiovascular mortality in elderly Chinese [11]. However, Li’s study included participants with 0.9 < ABI ≤ 1.3, and a more prolonged UT was shown in the non-survivor group as compared with that in the survivor group, but the difference did not reach statistical significance in the multivariate analysis. In the same study, they found that %MAP > 45% was a useful predictor of total mortality after multivariate analysis [8]. In the present study, we similarly found that increased UT was associated with total and cardiovascular mortality in the univariate analysis, but such an association disappeared after multivariate analysis. Hence, calculation of UT might not provide additional benefit in survival prediction in HD patients.

Li et al. demonstrated that %MAP had a positive correlation with baPWV in patients with ABI > 0.9, but UT had no correlation with baPWV in the same group [8]. In the present study, we consistently found a similar finding in HD patients. Although %MAP and UT had a moderate correlation (r = 0.427, P <0.001), %MAP had a significantly positive correlation with baPWV but UT had no correlation with baPWV. Hence, increased arterial stiffness might have a role in the %MAP prolongation, but not in the UT prolongation. In addition, a significant correlation between %MAP and baPWV might partially explain why %MAP was a useful predictor of total mortality and %MAP > 50% was a useful predictor of total and cardiovascular mortality in the present study.

Several traditional cardiovascular risk factors, such as diabetes, coronary artery disease, stroke, and chronic heart failure were useful predictors of mortality in HD patients [19,20,21]. In the present study, the hazard ratios (HRs) of these traditional cardiovascular risk factors for the prediction of total mortality were between 2.153 and 4.392 in the univariate analysis. The HR of %MAP > 50% for the prediction of total mortality was 3.758 in the univariate analysis. Hence, compared to these traditional cardiovascular risk factors, %MAP > 50% had a similar impact on total mortality in our HD patients.

## 5. Study Limitation

There were several limitations in this study. First, the study generality was limited because study patients were only included from one dialysis clinic in a regional hospital in southern Taiwan. Second, we did not evaluate the association of peripheral arterial occlusion disease on image studies with %MAP and UT. Therefore, the mechanism underlying the link between high ankle %MAP and mortality and no correlation between ankle UT and mortality was still unknown. Third, vascular calcification, extremely high in dialysis patients, might have affected the measurements of baPWV, ABI, %MAP, and UT. However, we did not evaluate the vascular calcification in our study patients, so the impact of vascular calcification on these parameter measurements could not be assessed in the present study. Fourth, most of our patients were receiving long-term antihypertensive treatment. For ethical reasons, we did not withdraw these medications. Therefore, we could not exclude the impact of antihypertensive medication on the present findings. Finally, because of the large number of variables in the analysis with only 29 mortality events, the possibility of a chance finding and the limited power should be considered.

## 6. Conclusions

In HD patients, we found that ankle %MAP and ankle %MAP > 50% could predict total mortality and ankle %MAP > 50% could predict cardiovascular mortality in the multivariate analysis. Hence, assessment of %MAP from pulse volume recording of lower limbs might be helpful in identifying the high-risk group for poor prognosis in HD patients.

## Figures and Tables

**Figure 1 jcm-08-02045-f001:**
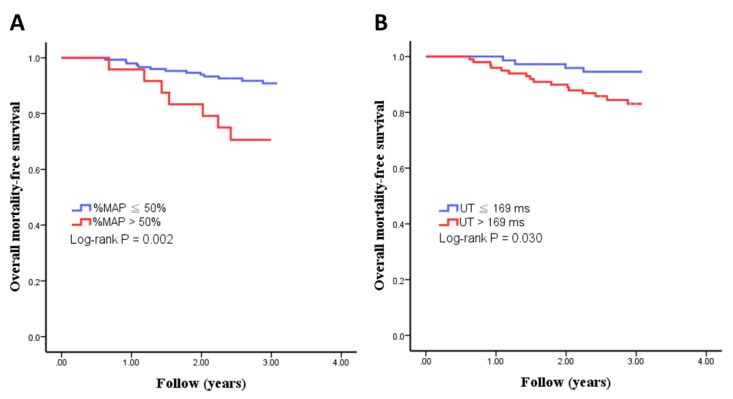
Kaplan–Meier curves for overall mortality-free survival in all study patients subdivided according to %MAP > 50% or not ((**A**): log-rank *p* = 0.002) and UT > 169 ms or not ((**B**): log-rank *p* = 0.030).

**Table 1 jcm-08-02045-t001:** Comparison of baseline characteristics between patients with and without mortality.

Characteristics	Patients with Mortality (*n* = 29)	Patients without Mortality (*n* = 168)	*p* Value	All Patients (*n* = 197)
Age (year)	68 ± 12	60 ± 11	<0.001	61 ± 12
Male gender (%)	52	54	0.854	53
Dialysis duration (month)	93 ± 61	91 ± 69	0.848	91 ± 68
Diabetes mellitus (%)	72	42	0.003	47
Hypertension (%)	59	51	0.424	52
Current smoking (%)	14	14	0.988	14
CAD (%)	28	7	0.001	10
Stroke (%)	24	7	0.004	10
CHF (%)	41	24	0.047	26
Fontaine’s stages III–IV (n)	5	20	0.425	25
SBP (mmHg)	162 ± 29	154 ± 27	0.234	155 ± 27
DBP (mmHg)	83 ± 18	82 ± 15	0.762	82 ± 15
MAP (mmHg)	109 ± 20	106 ± 18	0.450	106 ± 19
BMI (kg/m^2^)	22.6 ± 3.8	23.9 ± 3.8	0.103	23.7 ± 3.8
Albumin (g/dL)	3.6 ± 0.3	3.9 ± 0.3	<0.001	3.9 ± 0.3
Hemoglobin (g/dL)	10.3 ± 1.7	10.5 ± 1.1	0.350	10.5 ± 1.2
Total cholesterol (mg/dL)	161 ± 46	181 ± 38	0.013	178 ± 40
Triglyceride (mg/dL)	162 ± 141	181 ± 38	0.823	167 ± 123
**Medications**				
ACEI and/or ARB use (%)	28	21	0.462	22
β-blocker use (%)	24	20	0.633	21
CCB use (%)	38	23	0.093	25
**Peripheral vascular parameters**				
ABI in the lower side	0.87 ± 0.28	0.97 ± 0.20	0.039	0.96 ± 0.21
ABI < 0.9 in either leg (%)	48	26	0.041	27
ABI > 1.3 in either leg (%)	14	8	0.389	9
baPWV (cm/s)	1997 ± 763	1927 ± 505	0.586	1936 ± 537
UT (ms)	202 ± 57	82 ± 36	0.027	184 ± 39
UT > 169 ms (%)	80	55	0.031	58
%MAP	47.7% ± 5.8%	44.8% ± 4.5%	0.010	45.1% ± 4.8%
%MAP > 50% (%)	35	11	0.004	14

ABI: ankle–brachial index; ACEI: angiotensin-converting enzyme inhibitor; ARB: angiotensin II receptor blocker; baPWV: brachial–ankle pulse wave velocity; BMI: body mass index; CAD: coronary artery disease; CCB: calcium channel blocker; CHF: chronic heart failure; DBP: diastolic blood pressure; %MAP: percent of mean arterial pressure; SBP: systolic blood pressure; UT: upstroke time.

**Table 2 jcm-08-02045-t002:** Univariate and multivariate correlates of %MAP in study patients.

	Univariate Analysis		Multivariate Analysis		
	r	*p*	β	*p*	
Age (year)	0.228	0.003	−0.032	0.609	
Male gender (%)	−0.271	<0.001	−0.168	0.003	
Dialysis duration (month)	−0.018	0.812			
Diabetes mellitus (%)	0.327	<0.001	0.021	0.748	
Hypertension (%)	0.006	0.935			
Current smoking (%)	−0.132	0.083			
CAD (%)	0.078	0.31			
Stroke (%)	0.185	0.015	0.076	0.185	
CHF (%)	0.068	0.373			
SBP (mmHg)	0.249	0.001	0.121	0.054	
DBP (mmHg)	−0.057	0.464			
MAP (mmHg)	0.092	0.235			
BMI (kg/m^2^)	−0.158	0.038	−0.085	0.144	
Albumin (g/dL)	−0.229	0.002	−0.095	0.113	
Hemoglobin (g/dL)	0.027	0.725			
Total cholesterol (mg/dL)	0.118	0.122			
Triglyceride (mg/dL)	0.079	0.302			
**Medications**					
ACEI and/or ARB use (%)	0.022	0.776			
β-blocker use (%)	0.113	0.139			
CCB use (%)	−0.02	0.794			
**Peripheral vascular parameters**					
ABI in the lower side	−0.533	<0.001		−0.259	0.01
ABI < 0.9 in either leg (%)	0.457	<0.001		−0.035	0.699
ABI > 1.3 in either leg (%)	−0.59	0.443			
baPWV (cm/s)	0.290	<0.001		0.219	0.001
UT (ms)	0.615	<0.001		0.427	<0.001

r: Pearson correlation; β: unstandardized coefficient; other abbreviations as in Table 1.

**Table 3 jcm-08-02045-t003:** Univariate and multivariate correlates of UT in study patients.

	Univariate Analysis		Multivariate Analysis	
	r	*p*	β	*p*
Age (year)	0.187	0.014	−0.029	0.646
Male gender	0.112	0.145		
Dialysis duration (month)	0.003	0.972		
Diabetes mellitus	0.322	<0.001	0.025	0.7
Hypertension	0.014	0.853		
Current smoking	−0.059	0.441		
CAD	0.217	0.004	0.094	0.125
Stroke	0.096	0.209		
CHF	0.184	0.016	0.07	0.254
SBP (mmHg)	0.058	0.457		
DBP (mmHg)	−0.173	0.024	−0.143	0.017
MAP (mmHg)	−0.068	0.382		
BMI (kg/m^2^)	−0.110	0.152		
Albumin (g/dL)	−0.129	0.092		
Hemoglobin (g/dL)	0.083	0.277		
Total cholesterol (mg/dL)	0.054	0.481		
Triglyceride (mg/dL)	0.08	0.297		
**Medications**				
ACEI and/or ARB use	0.2	0.009	0.094	0.135
β-blocker use	0.213	0.005	0.068	0.284
CCB use (%)	0.083	0.277		
**Peripheral vascular parameters**				
ABI in the lower side	−0.520	<0.001	−0.216	0.035
ABI < 0.9 in either leg (%)	0.497	<0.001	0.090	0.350
ABI > 1.3 in either leg (%)	−0.053	0.487		
baPWV (cm/s)	−0.001	0.992		
%MAP	0.615	<0.001	0.427	<0.001

r: Pearson correlation; β: unstandardized coefficient; other abbreviations as in Table 1.

**Table 4 jcm-08-02045-t004:** Predictors of total and cardiovascular mortality using the Cox proportional hazards model in the univariate analysis.

Parameter	Total Mortality		Cardiovascular Mortality	
	HR (95% CI)	*p*	HR (95% CI)	*p*
Age (year)	1.070 (1.034–1.107)	<0.001	1.059 (1.007–1.112)	0.025
Male gender	0.947 (0.457–1.962)	0.883	1.598 (0.535–4.768)	0.397
Dialysis duration (month)	1.000 (0.995–1.006)	0.872	1.002 (0.994–1.009)	0.644
Diabetes mellitus	3.275 (1.449–7.339)	0.003	4.620 (1.287–16.585)	0.01
Hypertension	1.378 (0.658–2.887)	0.393	0.735 (0.255–2.119)	0.567
Current smoking	1.030 (0.358–2.959)	0.957	1.746 (0.487–6.260)	0.386
CAD	4.392 (1.939–9.950)	<0.001	6.421 (2.136–19.300)	<0.001
Stroke	3.774 (1.609–8.853)	0.001	4.758 (1.486–15.232)	0.004
CHF	2.153 (1.028–4.509)	0.037	2.285 (0.793–6.590)	0.116
Fontaine’s stages III–IV (%)	1.427 (0.544–3.740)	0.47	1.878 (0.524–6.732)	0.333
SBP (mmHg)	1.009 (0.994–1.025)	0.25	1.017 (0.995–1.039)	0.125
DBP (mmHg)	1.004 (0.977–1.032)	0.775	1.024 (0.989–1.061)	0.182
MAP (mmHg)	1.008 (0.986–1.032)	0.469	1.024 (0.993–1.056)	0.135
BMI (kg/m^2^)	0.909 (0.814–1.015)	0.094	0.935 (0.802–1.091)	0.397
Albumin (g/dL)	0.197 (0.088–0.441)	<0.001	0.221 (0.069–0.714)	0.013
Hemoglobin (g/dL)	0.859 (0.634–1.164)	0.33	0.859 (0.634–1.164)	0.33
Total cholesterol (mg/dL)	0.987 (0.977–0.997)	0.012	0.977 (0.962–0.992)	0.002
Triglyceride (mg/dL)	0.999 (0.996–1.003)	0.744	1.001 (0.997–1.004)	0.755
**Medications**				
ACEI and/or ARB use	1.482 (0.655–3.351)	0.342	2.199 (0.735–6.586)	0.148
β-blocker use	1.263 (0.539–2.958)	0.591	1.605 (0.503–5.125)	0.42
CCB use	1.895 (0.895–4.013)	0.089	1.723 (0.577–5.142)	0.324
**Peripheral vascular parameters**				
ABI in the lower side	0.151 (0.026–0.884)	0.035	0.205 (0.015–2.851)	0.237
ABI < 0.9 in either leg	2.375 (1.008–5.592)	0.041	1.763 (0.497–6.250)	0.373
ABI > 1.3 in either leg	1.730 (0.510–5.874)	0.379	2.623 (0.557–2.365)	0.223
baPWV (cm/s)	1.000 (0.999–1.001)	0.606	1.000 (1.000–021)	0.057
UT (ms)	1.010 (1.001–1.018)	0.027	1.013 (1.021–1.025)	0.022
UT > 169 ms	3.146 (1.051–9.411)	0.03	3.172 (0.674–14.941)	0.123
%MAP	1.121 (1.029–1.221)	0.009	1.106 (0.978–1.250)	0.108
%MAP > 50%	3.758 (1.497–9.434)	0.002	4.732 (1.333–16.800)	0.008

HR, hazard ratio; CI, confidence interval; other abbreviations as in Table 1.

**Table 5 jcm-08-02045-t005:** Predictors of total and cardiovascular mortality using the Cox proportional forward hazards model in the multivariate analysis.

Parameter	Total Mortality		Cardiovascular Mortality	
	HR (95% CI)	*p*	HR (95% CI)	*p*
UT (ms)	-	0.15	-	0.103
UT > 169 ms	-	0.184	-	0.387
%MAP	1.098 (1.001–1.204)	0.047	-	0.262
%MAP > 50%	2.900 (1.127–7.463)	0.012	4.295 (1.209–15.264)	0.024

HR, hazard ratio; CI, confidence interval; other abbreviations as in Table 1. Covariates in the multivariate model included significant variables in the univariate analysis in Table 4. Hence, we adjusted age, diabetes mellitus, CAD, stroke, CHF, albumin, total cholesterol, ABI, and ABI < 0.9 for total mortality and adjusted age, diabetes mellitus, CAD, stroke, albumin, and total cholesterol for CV mortality.

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
