# Peer review of "Association of Pulse Volume Recording at Ankle with Total and Cardiovascular Mortality in Hemodialysis Patients"

_jcm, 2019, doi:10.3390/jcm8122045_

Round 1

Reviewer 1 Report

The article submitted for the review is well prepared in terms of methodology, although there are few minor shortcomings:

1) Page 2: text should be rephrased "a group of extraordinarily high prevalence of PAD", because the "prevalence of PAD is high, ranging from 10 to 35%"

2) Page 5: The text should need further explanations for the data included in Figure 1

3) Page 8:  Checking the word "correlation": "but UT had no correlaiton with baPWV"

4) Page 8: the text should be rephrased "In addition, a significant correlation between %MAP and baPWV but not between UT and baPWV might partially explain %MAP, but not UT was a useful predictor of total mortality in the present study "

Author Response

Dear Editor:

Thank you for the thorough review on our manuscript (reference number: jcm-635612), entitled " Association of pulse volume recording at ankle with total and cardiovascular mortality in hemodialysis patients ". All the comments from the editor and reviewers are carefully considered, and the manuscript is revised according to the comments. We appreciate the reviewers’ kind instructions, suggestions, and corrections. For contrast, the corrections or additions are highlighted in red words in Microsoft Word.

Sincerely yours,

Ho-Ming Su, MD, E-mail: cobeshm@seed.net.tw

Cardiology/Internal Medicine, Kaohsiung Medical university, 100 Shih-Chuan 1st Road, Kaohsiung 807, Taiwan

Fax: (886) (7) 323-4845

The comments of the reviewers are as follows:

Review 1 comment

The article submitted for the review is well prepared in terms of methodology, although there are few minor shortcomings:

1) Page 2: text should be rephrased "a group of extraordinarily high prevalence of PAD", because the "prevalence of PAD is high, ranging from 10 to 35%"

--> Thanks for your great comment. Our manuscript was revised and rephrased in Text (page 1, line 40-41).

2) Page 5: The text should need further explanations for the data included in Figure 1

--> Thanks for your great comment. Further explanations in Figure 1 was added in our revised manuscript. Participants with %MAP > 50% and UT > 169 ms had higher total mortality rate. (page 5, line 146-147)

3) Page 8:  Checking the word "correlation": "but UT had no correlaiton with baPWV"

--> Thanks for your great comment. In our study, %MAP has a significant correlation with baPWV, but UT had no correlation with baPWV.

4) Page 8: the text should be rephrased "In addition, a significant correlation between %MAP and baPWV but not between UT and baPWV might partially explain %MAP, but not UT was a useful predictor of total mortality in the present study "

--> Thanks for your great comment. Our manuscript was revised and rephrased in Text (page 8, line 213-215).

Reviewer 2 Report

In this study, the authors assess the association between some arterial vessels parameters (MAP, UT, ABI) and all-cause and cardiovascular mortality in dialysis patients.

In my opinion the manuscript, despite some strengths including a large population and the rigorous methodological collection of study variables, has some concerns that need to be addressed.

Major comments

Despite stating in limitation that PAD was not diagnosed through Duplex or other traditional examinations, the number of patients with foot ulcers or resting pain (Fontaine's stages III-IV) need to be inserted in the manuscript.

In addition, in table 1 a single ABI value was reported, but what does value refer to? In my opinion, it could be the mean ABI value between the two limbs, but that value does not provide any scientific information. I suggest to insert both values (more impaired and less impaired limbs) in table, as well the number of patients with incompressible vessels (or ABI value >1.31) that is worldwide known as a higher predictor of CV mortality itself.

Results section contain a relevant bias that need to be discussed. Traditional cardiovascular risk factors including diabetes, CAD, CHF, stroke in Cox analyses has greater HR respect to UT and %MAP. That need to be discussed in the appropriate section.

In addition, how many patients exceeded the cutoff of MAP>50 and UT>169m/s? How did you choose that values? Are some references or did you arbitrarily choose that values? The low number of patients exceeding that values may explain the high HR observed in the Cox analyses.

Vascular calcification, extremely high in dialysis patients, may have affected all measures you present, including PWV, ABI, MAP and UT. This aspects need to be discussed or inserted in limitation section.

From a clinical point of view, what are the advantages of measuring MAP and UT respect to ABI only?

Minor comments 

Dialysis vintage should also be reported in tables and analyze as a possible factor affecting mortality.

The reference is good, but a brief statement for ABI and PWV measures in method section should be reported for the benefit of the reader.

There are several spelling mistakes, especially in discussion sections (lines 174, 188, 189); please correct.

In my opinion the terms univariate and multivariate analyses are preferable to univariable and multivariable.

Author Response

Dear Editor:

Thank you for the thorough review on our manuscript (reference number: jcm-635612), entitled " Association of pulse volume recording at ankle with total and cardiovascular mortality in hemodialysis patients ". All the comments from the editor and reviewers are carefully considered, and the manuscript is revised according to the comments. We appreciate the reviewers’ kind instructions, suggestions, and corrections. For contrast, the corrections or additions are highlighted in red words in Microsoft Word.

Sincerely yours,

Ho-Ming Su, MD, E-mail: cobeshm@seed.net.tw

Cardiology/Internal Medicine, Kaohsiung Medical university, 100 Shih-Chuan 1st Road, Kaohsiung 807, Taiwan

Fax: (886) (7) 323-4845

The comments of the reviewer 2 are as follows:

Review 2 comment

In this study, the authors assess the association between some arterial vessels parameters (MAP, UT, ABI) and all-cause and cardiovascular mortality in dialysis patients.

In my opinion the manuscript, despite some strengths including a large population and the rigorous methodological collection of study variables, has some concerns that need to be addressed.

Major comments

1) Despite stating in limitation that PAD was not diagnosed through Duplex or other traditional examinations, the number of patients with foot ulcers or resting pain (Fontaine's stages III-IV) need to be inserted in the manuscript.

--> Thanks for your great comment. Patients with resting pain or foot ulcers were 13% (n= 25) in our participants. Fontaine's stages III-IV did not impact on total mortality and CV mortality. We added that in revised manuscript and Tables (page 3, table 1; page 6, table 4)

2) In addition, in table 1 a single ABI value was reported, but what does value refer to? In my opinion, it could be the mean ABI value between the two limbs, but that value does not provide any scientific information. I suggest to insert both values (more impaired and less impaired limbs) in table, as well the number of patients with incompressible vessels (or ABI value >1.31) that is worldwide known as a higher predictor of CV mortality itself.

--> Thanks for your great comment. We all agreed and appreciated your comments that mean ABI value between the two limbs did not provide any scientific information. After obtaining bilateral values, we chose the lower ABI for later analysis. In addition, we added ABI > 1.3 in either leg in revised manuscript and Tables (page 3, table 1; page 4, table 2; page 5, table 3; page 6, table 4)

3) Results section contain a relevant bias that need to be discussed. Traditional cardiovascular risk factors including diabetes, CAD, CHF, stroke in Cox analyses has greater HR respect to UT and %MAP. That need to be discussed in the appropriate section.

--> Thanks for your great comment and very important point. Several traditional cardiovascular risk factors, such as diabetes, coronary artery disease, stroke, and chronic heart failure were useful predictors of mortality in HD patients [19-21]. In the present study, the hazard ratios (HRs) of these traditional cardiovascular risk factors for prediction of total mortality were between 2.153-4.392 in the univariate analysis. The HR of %MAP > 50% for prediction of total mortality was 3.758 in the univariate analysis. Hence, compared to these traditional cardiovascular risk factors, %MAP > 50% had a similar impact on total mortality in our HD patients. (page 8, line: 216-221)

4) In addition, how many patients exceeded the cutoff of MAP>50 and UT>169m/s? How did you choose that values? Are some references or did you arbitrarily choose that values? The low number of patients exceeding that values may explain the high HR observed in the Cox analyses.

--> Thanks for your great comment. There were 24 and 99 patients exceeded the cutoff value of %MAP > 50% and UT >169 ms. Due to limited optimal reference for cutoff values of %MAP and UT, we found the suitable cutoff valve by statistical methods. We created several models using different cut-off values of %MAP and UT. Using the Chi-square value to select the model with the best performance, the model using %MAP > 50% and UT > 169 ms had the best performance in predicting the overall mortality. (page 5, line: 139-147)

5) Vascular calcification, extremely high in dialysis patients, may have affected all measures you present, including PWV, ABI, MAP and UT. This aspects need to be discussed or inserted in limitation section.

--> Thanks for your great comment and outstanding view point. Vascular calcification, extremely high in dialysis patients, might have affected the measurements of baPWV, ABI, %MAP, and UT. However, we did not evaluate the vascular calcification in our study patients, so the impact of vascular calcification on these parameter measurements could not be assessed in the present study. (page 8, line: 227-231)

6) From a clinical point of view, what are the advantages of measuring MAP and UT respect to ABI only?

--> Thanks for your great comment. Our present study demonstrated even after adjusting ABI and ABI < 0.9, increased %MAP was still associated with total mortality and %MAP > 50% was still associated with total and cardiovascular mortality in patients with HD. Therefore, additional assessment of ankle %MAP might be helpful in identification of HD patients with high mortality. (page 7, line: 194-197)

Minor comments 

1) Dialysis vintage should also be reported in tables and analyze as a possible factor affecting mortality.

--> Thanks for your great comment. The overall dialysis duration (months) was 91 ± 68. We found no significant association of dialysis duration with %MAT, UT, and mortality events. We added the variable in the revised manuscript and tables. (page 3, table 1; page 4, table 2; page 5, table 3)

2) The reference is good, but a brief statement for ABI and PWV measures in method section should be reported for the benefit of the reader.

--> Thanks for your great comment. We added a brief statement of ABI and baPWV measures in revised manuscript. In brief, the ABI was counted by the ankle systolic blood pressure divided by the higher arm systolic blood pressure. The baPWV was calculated as the transmitted distance from the pulse wave from brachial to tibial arteries divided by passage time of the pule wave. (page 2, line: 74-77)

3) There are several spelling mistakes, especially in discussion sections (lines 174, 188, 189); please correct.

--> Thanks for your great comment. We corrected typing mistakes in Discussion section.

4) In my opinion the terms univariate and multivariate analyses are preferable to univariable and multivariable.

--> Thanks for your great comment. We use the terms of univariate and multivariate in the revised

Round 2

Reviewer 2 Report

Authors should correct the number of PAD patients in table 1 at stages III-IV, writing them as n = instead of %.

Thanks

Author Response

Dear Editor:

Thank you for the thorough review on our manuscript (reference number: jcm-635612), entitled " Association of pulse volume recording at ankle with total and cardiovascular mortality in hemodialysis patients ". All the comments from the editor and reviewers are carefully considered, and the manuscript is revised according to the comments. We appreciate the reviewers’ kind instructions, suggestions, and corrections. For contrast, the corrections or additions are highlighted in red words in Microsoft Word.

Sincerely yours,

Ho-Ming Su, MD, E-mail: cobeshm@seed.net.tw

Cardiology/Internal Medicine, Kaohsiung Medical university, 100 Shih-Chuan 1st Road, Kaohsiung 807, Taiwan

Fax: (886) (7) 323-4845

The comment of the reviewer 2 is as follows:

Review 2 comment

Authors should correct the number of PAD patients in table 1 at stages III-IV, writing them as n = instead of %.

--> Thanks for your great comment. We use the number of PAD patients at stages III-IV in Table 1 in the revised (Page 3, Table 1)
